# Spatial Analysis of Dengue Clusters at Department, Municipality and Local Scales in the Southwest of Colombia, 2014–2019

**DOI:** 10.3390/tropicalmed8050262

**Published:** 2023-05-02

**Authors:** Catalina Marceló-Díaz, María Camila Lesmes, Erika Santamaría, José Alejandro Salamanca, Patricia Fuya, Horacio Cadena, Paola Muñoz-Laiton, Carlos Andrés Morales

**Affiliations:** 1Grupo de Entomología, Instituto Nacional de Salud, Bogotá 111321, Colombia; malesmes@udca.edu.co (M.C.L.); esantamaria@ins.gov.co (E.S.); pfuya@ins.gov.co (P.F.); paola.munoz1@udea.edu.co (P.M.-L.); 2Facultad de Ciencias Ambientales y de la Sostenibilidad, Programa de Ingeniería Geográfica y Ambiental, Universidad de Ciencias Aplicadas y Ambientales, UDCA, Bogotá 111166, Colombia; geogenoma2014@gmail.com; 3Programa de Estudio y Control de Enfermedades Tropicales, PECET, Universidad de Antioquia, Medellín 050010, Colombia; horaciocadena@gmail.com; 4Secretaría de Salud Departamental del Cauca, Popayán 190001, Colombia; carlos.morales@cauca.gov.co

**Keywords:** *Aedes aegypti*, Getis-Ord Gi*, Poisson regression, pupae index, spatio-temporal analysis

## Abstract

Dengue is an arbovirus transmitted by mosquitoes of the genus *Aedes* and is one of the 15 main public health problems in the world, including Colombia. Where limited financial resources create a problem for management, there is a need for the department to prioritize target areas for public health implementation. This study focuses on a spatio-temporal analysis to determine the targeted area to manage the public health problems related to dengue cases. To this end, three phases at three different scales were carried out. First, for the departmental scale, four risk clusters were identified in Cauca (RR ≥ 1.49) using the Poisson model, and three clusters were identified through Getis-Ord Gi* hotspots analysis; among them, Patía municipality presented significantly high incidence rates in the time window (2014–2018). Second, on the municipality scale, altitude and minimum temperature were observed to be more relevant than precipitation; considering posterior means, no spatial autocorrelation for the Markov Chain Monte Carlo was found (Moran test ˂ 1.0), and convergence was reached for b_1_–b_105_ with 20,000 iterations. Finally, on the local scale, a clustered pattern was observed for dengue cases distribution (nearest neighbour index, NNI = 0.202819) and the accumulated number of pupae (G = 0.70007). Two neighbourhoods showed higher concentrations of both epidemiological and entomological hotspots. In conclusion, the municipality of Patía is in an operational scenario of a high transmission of dengue.

## 1. Introduction

Dengue is the most rapidly expanding arboviruses in the world, and its incidence has increased in recent decades in tropical and subtropical countries, becoming a global public health problem [1,2]. In the Americas there was recorded the highest number of dengue cases in history in 2019 with more than 2.7 million cases, including 22,127 serious cases and 1206 deaths [3]. 

In Colombia, approximately 58% of the population is at risk of having the disease, since it is endemic to approximately 909 municipalities, and 108 municipalities are centres of dengue transmission [1]. In 2021, during the epidemiological week 52, about 53,334 cases were recorded, of which 50% corresponded to dengue with warning signs, and 1.8% corresponded to severe dengue. Although the behaviour of the disease at the national level was within the expected range, the resulting mortality showed a significant increase. Some departments, including Cauca, are in a situation of alert [4], and Cauca itself reported 336 cases over the course of 2022. Between 2020 and 2021, this department was also under alert since its epidemiological behaviour exceeded the expected threshold [4,5].

The main vector of dengue in Colombia is the *Ae. aegypti* mosquito, an anthropophilic species that is found mainly in urban areas. In general, this vector infests natural or artificial containers in or around homes [6]. Its distribution has been reported in all departments of Colombia up to 2302 m.a.s.l. [7]. Under optimal conditions of food availability and adequate oviposition sites, the average dispersal of a female *Aedes* spp. mosquito is estimated to be between 50 and 100 m, which limits its visits to two or three dwellings during its adult life [8], although fed females have been recorded to disperse as far as 800 m in 6 days [9].

The dynamics that allow the transmission of dengue respond to a series of general and specific factors known as macrodeterminants and microdeterminants of dengue. Macrodeterminants can be (1) environmental, such as altitude, latitude and climatic conditions that facilitate larval development, vector survival and virus replication in the vector [10] and (2) demographic and socioeconomic conditions, such as intraurban mobility, inadequate urbanization patterns and lack of access to water supply and sewerage services, as well as cultural conditions, such as a low perception of risk. Microdeterminants include host characteristics, the circulation of the various dengue serotypes and entomological factors. These considerations highlight the presence of various conditions that favour the endemic–epidemic transmission of dengue in the different municipalities of the department of Cauca [10].

These variables provide valuable information, which is collected by health institutions and thus forms the early warning components of systematic public health surveillance systems, when this surveillance is based on indicators. Although dengue surveillance is traditionally syndromic or sentinel, there are also alternative sources of information which do not necessarily belong to the health field and could be taken into account to carry out surveillance based on indicators such as meteorological, entomological and environmental data.

Given the multiplicity of factors that affect the transmission of dengue, geographic information systems (GISs) have become a very useful tool for identifying the distribution patterns of events and studying their association with the macrodeterminants mentioned above.

Thus, the use of global and local spatial-temporal analysis tools such as the nearest neighbour index (NNI) [11]; the Getis Ord-Gi* statistic, used to detect hot and cold dengue spots with geographically homogeneous high or low values [12]; and kernel density, applied in several studies on dengue [13,14] among other techniques, allow the generation of maps that highlight the geographic areas and population groups at risk [15,16]. Those generated maps can serve as a basis for responsible institutions to design strategies for preventing and reducing the risk of dengue epidemics. 

In order to assist in determining the problematic dengue areas to ensure an effective management of the dengue cases, a spatio-temporal analysis was conducted. This analysis contributes to the process of spatial stratification, allowing the prioritization of control actions, to generate base information for developing contingency plans for vector-borne diseases, such as dengue fever.

The primary objective of this study was to analyse the spatio-temporal distribution of dengue in the department of Cauca to identify high-dengue-risk municipalities through spatial scan statistics and cluster analysis. 

The department of Cauca was selected due to its epidemiological behaviour in the last ten years, with the most important epidemic outbreak recorded in this period occurring between the end of 2009 and 2010. The endemic–epidemic characteristics of Cauca provide a suitable study area to carry out a series of spatial analyses since it presents temporal, seasonal, and cyclical behaviours in at-risk populations, enabling a better characterization without presenting biases due to epidemics or hyperendemic trends.

A secondary aim was to contribute to the spatial stratification of dengue in the identified high-risk municipalities, which includes determining the effect of environmental risk factors on the rate of dengue cases and the identification of important transmission foci [8]. With this study, the finding determines the level of risks and helps to determine actions needed to control the risks.

A third objective was to prioritize vector-control activities through the identification and analysis of persistent dengue and entomological clusters by neighbourhood, to reveal correlations and to generate base information for the elaboration of contingency plans for vector-borne diseases, such as dengue, in problem areas which will be targeted for intensive intervention.

## 2. Materials and Methods

The study comprised three phases. The first phase targeted the locations of risk clusters in the department of Cauca through a spatio-temporal analysis. The department of Cauca was selected because it has been on epidemiological warning for more than three consecutive years since 2019, given the higher-than-expected increase in dengue cases, as reported in the epidemiological reports of the National Institute of Health (INS, acronym in Spanish) [4,5]. In the second phase, environmental variables related to the disease were identified through the development of a Poisson regression model for the municipality of Patía, which was selected because it shows a relatively moderate dengue risk. In the third phase of the study, areas with higher-than-expected incidence of the disease were established and clusters were identified at the neighbourhood scale (an urban spatial unit), using spatial analysis processes in GIS software. These urban spatial units should be targeted for intensive intervention and surveillance. Figure 1 shows the development of each of the phases.

### 2.1. Study Area

The department of Cauca is located in southwestern Colombia between the coordinates 00°58′ and 3°19′ north latitude and 75°47′ and 77°57′ west longitude. It is divided into 42 municipalities covering an area of 29,308 km^2^ and has a population of 1,491,937 inhabitants [17]. Its average altitude is 1693 m.a.s.l.; however, it contains all thermal floors, which is reflected in its diversity of ecosystems. It contains mangrove forests; moors; wetlands; warm forests; sub-Andean, Andean and high-Andean forests; inter-Andean valleys; and dry and subxerophytic forests [18].

The climate in the department of Cauca has a great variability depending on its geographic and geomorphologic characteristics. On the Pacific coast of the department, there are warm wet and semi-humid pluvial climates, with rainfall that can range from 3000 mm to 7000 mm per year and average temperatures of 24 °C. In the central subregion, the climate is predominantly temperate wet, with temperatures that oscillate between 18 °C and 24 °C. The Patía Valley stands out as the driest area in the department as it has a warm dry climate and receives less than 1500 mm of rainfall per year. In contrast, the eastern edge of Cauca has a cold dry and very cold dry climate, while the Amazonian foothills has a warm wet climate.

Cauca has seven subregions: the central subregion, where the capital Popayán is located; the northern subregion, divided into 13 municipalities, including Miranda, Puerto Tejada and Villa Rica; the eastern subregion; the Pacific subregion; the southern subregion, where the municipality of Patía (see Section 2.3.2) and a large part of the Patía River valley are located; the subregion of the Massif; and the Amazonian Piedmont subregion, which includes the municipality of Piamonte [18].

For each phase of the study, we considered the results obtained in the previous phase. For Phase 1, Cauca was chosen as we explained before. For Phase 2, Patía municipality was selected considering the findings of spatio-temporal analysis and for Phase 3 we chose its municipal seat, because of its important as an administrative centre, seat of government, or capital city. Additionally, this municipal seat has been positive for pupae sampling throughout several years in several neighbourhoods.

### 2.2. Data Description

Epidemiological data regarding dengue and severe dengue cases were used in each of the phases of the study, including laboratory-confirmed cases and cases confirmed by epidemiological linkage. Data also included sociodemographic characteristics at the patient level such as age, sex, occupation, and ethnic data. These data are routinely reported by the Unidades Primarias Generadoras de Datos (UPGD in spanish) following the guidelines established by the WHO in 2009 [19].

In Colombia, a laboratory-confirmed case is a probable case of dengue, severe dengue, or dengue mortality confirmed by any of the laboratory criteria for dengue diagnosis. A positive PCR or viral isolation in patients with less than 5 days of onset of fever or Dengue IgM Dengue ELISA in patients with 5 or more days of onset of fever, while a confirmed case by epidemiological linkage involves confirming probable cases of dengue from laboratory-confirmed cases using the association of person, time and space [19]. This information is recorded in a database that contains epidemiological records by epidemiological week, so for the present study it was necessary to group them by year. 

The procedures and specifications applied to the data for the development of the phases of the study are described below:

#### 2.2.1. Dengue Incidence Data for the Department Scale

For the spatio-temporal analysis, dengue event notification data from the department of Cauca recorded in the period 2012 to 2018, acquired through the National System of Public Health Surveillance (SIVIGILA, for its Spanish abbreviation) were used. Data available at https://www.ins.gov.co/Direcciones/Vigilancia/Paginas/SIVIGILA.aspx (accessed on 10 January 2020).

The year 2019 was excluded from the analysis because it was an epidemic period; therefore, to avoid overestimating the cluster, that year was not considered in the spatio-temporal analysis. For hot spots detection, the incidence rates of dengue per 10,000 inhabitants from 2014 to 2018 were calculated from the cases reported to the SIVIGILA, and the population statistics of the projection system reported by the National Administrative Department of Statistics in 2020; in both process data were georeferenced. In 2014, a total of 378 cases of dengue fever were registered in the department of Cauca, with an increase of 40% in 2015 with 631 cases. About 803 cases were confirmed in 2016 and only 96 cases in 2017. Finally, in 2018, there were 177 cases registered. 

#### 2.2.2. Dengue Data for the Local Scale

For the municipality of Patía the databases of dengue and severe dengue cases from 2015 to 2019 reported by the Epidemiology Surveillance Group of the Department of Health of Cauca were reviewed (data available by request). Of the 246 total cases, 66 were discarded for reasons such as duplication of information, addresses that could not be found or different origins. As a result, 180 cases of dengue were used, including six cases of severe dengue, of whom 167 were georeferenced and 13 were geocode. To explore the data, a dengue endemic channel for the municipality of Patía was developed. An endemic channel represents the number of cases within the expected normal seasonal range; anything above this moving threshold would be considered representative of an unprecedented number of cases, i.e., an outbreak (Figure 2). The zones of success, security, and alarm determine the habitual behavior of the dengue cases.

Events were differentiated when they had been confirmed by a laboratory or clinic, confirmed by an epidemiological linkage, discarded, updated, or adjusted due to typing error. Subsequently, only the variables that were necessary to carry out the geocoding or georeferencing process were selected as appropriate. Whether the case had moved from his or her municipality of residence was also noted, and the location of the address and/or the neighbourhood was taken into account.

The geocoding process, which consists of assigning geographic coordinates to the addresses, was carried out in cases where the detailed description of the location was available, through Google Maps Geocoding API [20]. Georeferencing, which also involves the positioning of the element in a coordinate system, was applied for general descriptions referring to polygon-type elements. 

Georeferencing was carried out using the point-radius method [21]. The assignment of coordinates was performed using the centroid of the most specific political divisions, which could correspond to a village, section, or neighbourhood. This centroid was calculated using the Calculate Geometry tool of ArcGIS^®^ 10.8 for regular polygons and the Point on Surface tool of QGIS 3.10.5 for irregular polygons, using the official layers of the rural and urban sections provided by DANE [22]. For the polygons of neighbourhoods in the municipality, the layer was downloaded from the collaborative project Open Street Maps (OSM) due to its greater precision (https://www.openstreetmap.org/#map=5/4.632/-74.299 accessed on 25 June 2020). Data with ambiguous locations and data with neighbourhood or village locations that did not correspond to the information of the municipality or cases that reported a municipality of residence other than Patía were excluded.

### 2.3. Study Design

A retrospective study based on the epidemiological data collected for the observation window from 2012 to 2019; after performing its georeferencing, the study was developed at three scales: departmental, municipal, and local. 

#### 2.3.1. Phase 1: Spatio-Temporal Analysis of Dengue Distribution for Cluster Detection, Targeting High-Dengue-Risk Clusters of Municipalities at the Departmental Scale (Objective 1)

Space-time clustering: A space-time scan statistic for the detection of risk clusters following the Poisson model [23] was used for the years 2012 to 2018 when a spatial pattern of dengue cases existed. Targeting was carried out using the reported epidemiological information and included the following steps: (1) grouping of the epidemiological cases of the event reported to SIVIGILA according to the municipality of residence of the case or where the case originated, (2) information regarding the total number of cases of dengue and severe dengue taking into account the time of year and (3) location of the centroids in terms of latitude and longitude for each municipality in the department.The information was processed using Kulldorff’s method [23]. This method uses a model in which the number of events in a geographic area is distributed according to the Poisson model; under the null hypothesis, and without covariates, the expected number of cases in each area is proportional to its population size, or to the person-years in that area. Because the data are aggregated in census districts, the measurement was concentrated in terms of the central coordinates of those districts and was expanded along a third dimension that reflects the size of the population as it changes over time.The retrospective space-time analysis scanning for clusters with high rates using the discrete Poisson model was defined by a cylindrical window with a circular geographic base and with a height corresponding to time. The base was defined exactly as for the purely spatial scan statistic, while the height reflects the time frame of potential clusters. The cylindrical window was then moved in space and time, so that for each possible geographical location and size, it also visited each possible time frame. In effect, we obtained an infinite number of overlapping cylinders of different size and shape, jointly covering the entire study region, where each cylinder reflected a possible cluster [23].Hotspot detection: Once the overall landscape of the Cauca department was defined by clustering risk, a more detailed analysis was carried out to distinguish the year-to-year epidemiological behaviour of each municipality using the High-Low Clustering technique (Getis-Ord General and Getis Ord-Gi* statistic). This technique identifies the areas where the highest incidence rates of dengue are geographically homogeneous and concentrated under statistical significance parameters. These areas, known as hot and cold spots, as they specifically consider the extreme high or low values, have been previously used by authors such as Khormi and Kumar [24] and Mutheneni et al. [12] to study the spatial patterns of dengue and their potential application in disease management programmes.The clusters that presented a pattern of distribution of the incidence rates of dengue per 10,000 inhabitants—that is, data that were not randomly distributed in the department of Cauca for the period between 2014 and 2018—were identified. The conceptualization of inverse distance was determined using the Euclidean distance without defining a distance threshold. Additionally, the statistical significance was based on the False Discovery Rate (FDR) correction.These spatial geostatistical techniques allowed us to characterize the geographic distribution of the incidence of dengue [25] and to thereby identify statistically significant hot spots and cold spots using the Getis-Ord Gi* statistic. This statistic allowed us to examine the local level of spatial clustering to identify and visualize the municipalities whose dengue rate values were extreme and geographically homogeneous.

#### 2.3.2. Phase 2: Spatial Stratification of Dengue Risk Villages by Poisson Regression, to Determine the Effect of Environmental Factors That Influence the Spatial Variation of Dengue at the Municipal Scale (Objective 2) 

Study site: Patía municipality was chosen for spatial and environmental factors analysis within the department after carrying out Phase 1; the municipality had one of the highest incidences of dengue fever in the time window (2012–2018). In relation to entomological indicators, it has been positive for pupae index throughout the years in several neighbourhoods of its municipal seat.The municipality is located in the department of Cauca and has an area of 723 km^2^. As of 2021, it had a population of 37,793 inhabitants, of whom 13,598 lived in the urban area. The municipal seat, called Bordo-Patía, is located at the coordinates 02°06′56″ N and 76°59′21″ W and at an altitude of 910 m.a.s.l. The average temperature of the municipality is between 25 and 27 °C (maximum temperature 33–38 °C and minimum temperature 15–19 °C), and the average annual precipitation is 2171 mm. The municipality includes zones of premontane rainforest, premontane dry forest, and tropical dry forest [26].The climate in Patía is conditioned by its geoforms. A warm climate is representative of the depression or valley of Patía, which covers most of its extent. A cold climate is associated with the western mountain range that passes through the north of the municipality. Precipitation in the municipality has a bimodal distribution and is divided into two wet periods (March–May and October–December, the latter of which is more intense) separated by two dry periods (January–February and June–September) [27].Environmental factors: The environmental variables that were considered in the Poisson regression were altitude, minimum and maximum temperature, and precipitation. To acquire the data associated with these variables, the NASA Earth Data platform and the Climate Engine climate database were used.Altitude data were obtained from the NASA website, https://search.asf.alaska.edu/#/ (accessed on 8 October 2020), which has a variety of high-resolution cartographic resources. To download the digital elevation model (DEM) from which the altitude data were derived, a user account was created to access the platform. After the area of interest was located, the dataset “ALOS PALSAR” was selected, which contains the global DEM at a resolution of 12.5 m obtained through the synthetic aperture radar of the Advanced Land Observation Satellite (ALOS).The climate data were acquired through the website https://app.climateengine.org/climateEngine (accessed on 24 March 2021); after logging into the platform, a series of parameters were defined, such as the dataset, the meteorological variables of precipitation, maximum and minimum temperature, and the time interval (2015–2019). The dataset selected for the study was Terraclimate, which has monthly information on all required variables with a spatial resolution of 4 km. It is the most detailed among the various datasets available in the Climate Engine platform, which covers pixel values between 5 × 5 km and 55 × 55 km. Additionally, this dataset has an array of data from WorldClim, CRUTs 4.0 and JRA 55, which were structured and validated using interpolation and reanalysis techniques [28].Poisson regression analysis: Prior to modelling, univariate Poisson regression models were developed to identify the environmental factors that influence the increase of and spatial variation in the dengue burden in the municipality; a Poisson regression model was run using ArcGis Pro© software, taking into consideration the statistically significant variables between altitude, temperature (maximum and minimum) and precipitation.To identify how the environmental factors influence the increase of and spatial variation in the dengue burden in a village within the municipality, a Poisson regression model was run using WinBUGS [29] and GeoBUGS [30] software packages, both useful for making inferences under a Bayesian framework using the Gibbs Sampling method.WinBUGS approaches Bayesian estimation problems by multiplying *a priori* distribution by the likelihood and then simulating samples from the *a posteriori* distribution using the Gibbs algorithm. Predictive maps for the risk of infection were obtained taking into consideration the statistically significant variables such as altitude and minimum temperature.Subsequently, a Hierarchical Bayesian Model (HBM) was built in two stages; the HBM uses multiple levels of analysis in an iterative way [31]. As described by Best [32], in the HBA the unexplained extra-variance found in spatial statistics is identified as either spatially correlated effects or heterogeneity effects.A purely spatial modelling in two stages was followed: at the first stage, a likelihood model for the observed and expected dengue disease counts was specified based on the environmental variables. At the second stage, a prior model over the space of possible relative risks (RR) was specified. The data included two covariates measuring the minimum temperature and elevation within the municipality, and a list of adjacent villages, using the intrinsic conditional autoregressive (CAR) prior proposed by Besag, York and Mollie [33]. This model considers the spatial correlation between neighbouring areas. The general model could be written as:

Oi ~ Poisson(μ_i_),log (µ_i_) = log E_i_ + α_0_ + α_1_x_i_/10 + b_i_,(1)

Where α_0_ is the log relative risk for dengue in the study region, x_i_ is one of the covariates with associated regression coefficient α_1_, and b_i_ is an area that represents the residual or unexplained relative risk of disease. To allow for spatial dependence between the random effects b_i_ in nearby areas, the car.normal distribution was used. The set of posterior means of relative risks was then used to create a map to visualize high- or low-risk segments.The Bayesian interpretation of probability allows (proper prior) probabilities to be assigned subjectively to random events, in accordance with the natural history of the disease. The Markov Chain Monte Carlo (MCMC) implementation ran a sampling chain for 20,000 iterations, and the first 1000 iterations were discarded as pre-convergence “burn-in” [34]. Spatial autocorrelation was evaluated using Moran’s Index in the residuals. After running the HBM, we use the package CODA and R2WinBUGS to analyse the outputs [29,30,31,32].

#### 2.3.3. Phase 3: Spatial Analysis for the Identification of Epidemiological and Entomological Clusters at the Local Scale, to Prioritize Vector-Control Activities (Objective 3)

Spatial analysis with Hot Spot analysis: Prior to hot spot analysis, the Getis-Ord General G statistic was used to identify significant risk clusters. The z-score and *p*-value are measures of statistical significance that lead to acceptance or rejection of the null hypothesis. For this technique, the null hypothesis states that the values associated with features are randomly distributed. For this analysis, a radius of 100 m was established as threshold distance within the neighbourhood area. This radius was selected based on the dispersal area of the vector [9,35,36] and the results of other similar studies [37,38,39,40]. The potential clusters with the highest burden of dengue per unit area were identified, based on the incidence rate in the neighbourhoods that permanently contribute to the cumulative burden of dengue cases in the municipal seat of Patía.Getis-Ord Gi* test for entomological data: The existence of spatial distribution patterns of the pupal stage of *Ae. aegypti* vector was evaluated through the study of entomological variables at the neighbourhood scale, such as the total number of pupae, the index of pupae per person and the *Breteau* index (BI) for spatial analysis in GIS software. Those indicators were selected because the sub-department of health of Cauca (Secretaría de Salud) collected systematically and continuously in conducting this type of entomological surveillance.First, an analysis of the spatial distribution of the vector was performed, with a measure of the degree of clustering (high or low) using the general G statistic of Getis-Ord to indicate whether the pattern was uniformly or randomly clustered based on the fixed distance band parameter. Subsequently, neighbourhoods with groupings or high and homogeneous concentrations of clusters were identified with the Getis-Ord Gi* statistic using 100 m as a distance threshold. The statistical significance was based on the False Discovery Rate (FDR) correction. The information on the variables used was obtained through the Basic Entomology Unit of the Department of Health of Cauca (data available by request).For both spatial analyses we consider “local” in the sense of considering neighbourhood areas within the municipality.

### 2.4. Ethics Statement

Access to patient data, including the home addresses of the patients, was approved by the Ethics Committee of the National Institute of Health, Colombia (CEMIN 13-2019).

## 3. Results

### 3.1. Geographic Distribution of Dengue Incidence

Between 2012 and 2018, the Cauca department showed about 3023 dengue cases throughout the territory. A time-series plot of dengue virus cases (DENV,) in Figure 3, shows dengue cases on the vertical axis, and the time period for the analysis on the horizontal axis. The series shown below is non-stationary and non-linear, with trends that could be associated with a seasonal component. 

Figure 4 shows the variation of the incidence rate of dengue for the 42 municipalities of the department of Cauca between 2014 and 2018. Global cluster analysis was performed to represent a variable range between 0.10 and 83.23 cases per 10,000 inhabitants.

In general, the department of Cauca had high incidence rates of dengue in 2014, 2015 and 2018. The epidemiological behaviour of the department was characterized by having 376 cases in 2014; a 64% increase in 2015, with a cumulative incidence rate of 47.04 per 100,000 inhabitants; a gradual decrease in the incidence rate in 2016 and 2017 because of the effect of interventions in the Zika outbreak; and an unusual increase in cases in 2018, with a total of 177 accumulated cases (Figure 2 and Figure 3).

The municipalities in the northeast of the department (Miranda, Puerto Tejada, Villa Rica, Padilla, Corinto, Caloto and Santander de Quilichao) had high incidence rates every year, while the municipalities towards the Pacific coast (Guapi, Timbiquí and Algeria, considering that the latter had very low entomological indices of pupae) had variable and non-persistent incidence rates. Towards the southeastern part of the department, there was an unusual increase in the number of cases in Piamonte (IR = 75.3; POP = 9335), while in the southern region of the department, the municipality of Sucre had the highest incidence rate (IR = 83.23; POP = 9748). The municipality of Patía had continuously moderate to high incidence rates during the observation window (IR = 3.8–28.4; POP = 37,793).

#### 3.1.1. Identification of Clusters of High Risk for Dengue in Cauca

In the retrospective spatio-temporal analysis, the search for clusters with high rates using the discrete Poisson model allowed the identification of four statistically significant geographic clusters with a total of 2697 cases and an annual rate of 29.3 cases per 100,000 inhabitants. This model is useful when count data are available, where there is a background population from which the cases arise, and under the null hypothesis that the cases of dengue are independent of each other.

The first cluster was obtained for 2018, with the highest relative risk of 26.25, while the second cluster, which included the municipality of Patía, presented a relative risk of 8.17 (*p* ˂ 0.05) specifically for the period between January 2014 and December 2016. In this time frame, the observed cases (*n* = 293) were higher than the expected cases (*n* = 39.65) for these municipalities. The third cluster had a relative risk of 4.83 between 2013 and 2015 and was statistically significant, while the fourth cluster presented a value of 1.49 for 2016 (*p* > 0.1). Table 1 shows the results for each of the clusters from the analysis of the number of registered cases relative to the number of expected cases. A graphical representation of the four clusters is shown in Figure 5. 

#### 3.1.2. Getis-Ord Gi* Dengue Hotspot Detection

When analysing the spatial distribution of the incidence rate of dengue for the years 2014 to 2018 in the department, a hot spot analysis identified significantly high rates (≥95% confidence) in the municipalities of Miranda and Patía (year 2014), and the municipalities of Patía and Puerto Tejada (year 2015). No municipalities were identified as “cold” areas where the incidence rate was significantly low.

In the global analysis for 2014 and 2015, the presence of clusters was evidenced for the variable incidence rate (*p* < 0.1; z-score ≥ 1.65). The spatial pattern of the disease was distributed in an aggregated way that formed clusters, as the spatial analysis looks for patterns and meanings. In Figure 6, the dark and light red colours indicate hotspots of dengue cases (z-score Getis-Ord > 2.58 statistically significant). The blue and light blue colours represent cold spot areas (z-score Getis-Ord < 2.58 statistically significant). 

The presence of the municipality of Patía in the cluster analysis persisted until 2017, when it presented significantly high incidence rates, along with the municipality of Piamonte. However, the risk clusters identified in that year were not statistically significant (*p* > 0.1; z-score: 0.023974). For 2018, the municipalities of Piamonte and Santa Rosa showed high incidence rates, although the presence of *Ae. aegypti* was not reported in the municipal seat of Santa Rosa. This difference was not statistically significant (*p* > 0.1; z-score ˂ 1.65).

Once the geographical variation of the event was established, the areas within the municipality with a higher than expected incidence of the disease at the section and neighbourhood scales were identified. This analysis confirmed geographic variation with respect to the incidence rate variable, which was evidenced by identifying the neighbourhoods with higher or lower proportions of events.

### 3.2. Spatial Variation of the Probability of Dengue Incidence as a Function of Environmental Variables in the Municipality of Patía

In the epidemiological targeting for Patía, the municipality reached significant incidence rates of dengue in 2014, 2015, 2016 and 2018. During the latter year, the incidence rate was 6.5 per 10,000 inhabitants, with a total of 24 accumulated cases (Figure 2 and Figure 3). In 2015, 41% (*n* = 72) of the total number of cases in the period were analysed; meanwhile, 2016 gathered 32% (*n* = 56), followed by 2018 and 2019 with 11% (*n* = 19) and 10% (*n* = 17) respectively; the year with the lowest number of cases was 2017, with 7% (*n* = 12). Figure 7 shows the location of dengue cases registered in the period 2015–2019.

These incidence fluctuations may reveal a relationship between dengue disease and the explanatory environmental variables, increasing the probability of finding dengue and its vectors in the villages and population centres analysed, which will allow the carrying out of a characterization process, which is secondary to the targeting process and is useful for identifying and describing the main epidemiological and environmental variables that shape the dynamics of dengue transmission in one of the prioritized clusters of the disease at the departmental scale.

Using a count (Poisson) model type for our dependent discrete variable allowed us to identify the elevation (β = 0.0197) and the minimum temperature (β = 45,185) data as statistically significant (*p* ≤ 0.01). This results indicate that, in the municipality of Patía, dengue cases are positively influenced by altitude and minimum temperature. Both variables had a significant z-statistic (elev. = 11.2427; min.temp. = 10.4163), meaning that they could contribute significantly to the model. 

The identification of risky villages or towns within the municipality requires spatial analysis using GIS and Hierarchical Bayesian Modelling (HBM). It is common knowledge among vector-borne disease researchers that elevation is a risky factor for dengue, but the relationship between the number of cases and minimum temperature may still exist within each particular region. Figure 8 shows the changes in the relative risk (mean) of dengue infection through a HBM as a function of environmental variables: altitude and minimum temperature. This posterior relative risk map clearly shows the characteristic Bayesian smoothing of the crude relative risks. 

Additionally, the Caldas-Lang Climate Classification shown in Figure 8 is a classification method widely used in Colombia to characterize the climate, since it allows observation of the general behaviour of temperature as a function of the altitude and humidity of a certain region. This figure shows the risk levels divided into seven levels from lower to higher; a high probability of dengue disease is expected in 18 of 104 villages, which are located towards the northwest of the municipality, and in some isolated villages such as Piedra-Sentada, el Estrecho and El Bordo. 

The results of the characterization process show that although the probability of dengue infection for the years 2015 and 2019 as a function of the minimum temperature and altitude variables was higher in the northwest part of the department, some population centres, including El Bordo-Patía, were influenced by other variables, such as biological and sociodemographic variables. Table 2 shows the posterior means and posterior standard deviations of the Poisson regression coefficients related to dengue cases. The model run on WinBUGS software with 20,000 iterations starting at 1001.

From the relative risk estimations displayed in Table 2, analysis reveals that RR estimations at the northwest of the municipality are greater than those near the Patía River valley. The data displayed provide a picture about the application of the spatial model, such as BYM model, that is even more realistic than that the frequentist approach.

The spatial autocorrelation was evaluated with the regression residuals to assess if the Poisson model was correctly specified. Figure 9 shows the cross-correlations between variables in Markov Chain Monte Carlo output showing no spatial autocorrelation for the Markov chain (Moran Test ˂ 1.0). Trace and kernel density for priors (Figure 10) are both available as separate plots, but they are available together via the plot method using the CODA package through R. The plot shows convergence in the case of parameter b_1_ as well as the other parameters of b. However, the lack of convergence in the case of α_2_, specifically between the range of 10,000 and 15,000 iterations, may be a wrong sign about convergence if one takes into account that the density distribution is rather irregular. Data used for cross-correlations and trace and kernel density for nodes are contained within the Appendix A section.

The BYM-Poisson model is a good hierarchical distribution to model our data because of the nature of the outcome vatiable. Hence, one of the advantages of this model allows the identification of spatially aggregated count data when incidence is not so high. There are several methodological alternatives for generating estimates with Bayesian hierarchical models and the accuracy of the risk map could be increased for more detailed comparisons by including more explanatory variables. The BYM-Poisson model indicates that, in the municipality of Patía, dengue cases are positively and spatially influenced by altitude and minimum temperature. The results of the estimation can be used as the reference to anticipate the spread of dengue in a municipality, i.e., Patía. The strategic decision and action must be implemented in critical months and must be carried out by the ETV Programme and the Inspection, Surveillance and Health Control Process at Cauca to achieve a more optimal prevention.

### 3.3. Determination of Disease and Entomological Cluster in the Municipal Seat of Patía

Table 3 shows the result of the total data with the cases by origin, discarded cases, and those that were integrated into the geocoding or georeferencing process. Of the 246 cases used for assigning latitude and longitude, 66 cases were discarded for reasons such as duplication of information between neighbourhood and village, addresses that could not be found, or different origins. Of the total registered cases, 13% were displaced, and six of these cases were reported in Popayan as their municipality of residence and notification; this municipality is the capital city of Cauca (economic and administrative centre of the department).

In total, 180 cases of dengue, including six cases of severe dengue in the municipality of Patía that occurred between 2015 and 2019, were georeferenced (*n* = 167) and geocoded (*n* = 13). Of the 180 cases of dengue, 59.9% were men and 40.1% were women. The age range of those affected by the disease ranged from 1 to 89 years, with the following breakdown: 1–5 years, 6.2%; 6–11 years, 16.4%; 12–18 years, 15.8%; 19–26 years, 19.2%; 27–59 years, 35%; and 60 years and older, 7.3%.

After carrying out the global analysis, the −1.65 ≥ z-score ≤ 1.65 pattern did not appear to be significantly different than random at the local scale. Although the calculation of the Getis-Ord General G statistic (=0.001909) for the study period did not allow us to establish a clustering pattern (*p* < 0.1780; z-score: 1.3468), nine neighbourhoods were found to have more than 15 cases per 1000 habitants and four neighbourhoods more than 23 cases per 1000 habitants (Figure 11).

#### Spatial Pattern Analysis of Entomological Variables

Regarding the immature stage of the species, for the accumulated pupae for the years 2017 to 2019, the observed general G of Getis-Ord was greater than zero (G = 0.700066), showing that the variable has an aggregate/cluster distribution pattern. The null hypothesis test of complete spatial randomness (CSR) was rejected (Z coefficient = 1.808633, *p* < 0.1). This means that the spatial distribution of the high vector distribution values in the study area was more spatially clustered than would be expected if the underlying spatial distribution processes were random.

Figure 12 shows the neighbourhoods with statistically significant values. The analysis of hot spots by applying the Getis-Ord Gi* statistic to the 26 neighbourhoods showed that the Libertador neighbourhood has the highest spatial clustering pattern for the distribution of pupae applying the FDR correction (Gi *p*-value = 0.001; nNeighbors = 22). Additionally, when using the critical *p*-values and z-scores, the Modelo and Olaya Herrera neighbourhoods had a statistically significant pattern, with 90% confidence (*p* ≤ 0.1).

Figure 13 shows spatial autocorrelation for a series of distances and creates a line graph of those distances and their corresponding z-scores. Z-scores reflect the intensity of spatial clustering, and statistically significant peak z-scores indicate distances where spatial processes promoting clustering are most pronounced. For the accumulated pupae variable, we did not find peak distances statistically significant vis-à-vis the radius parameter established.

However, one peak at 536.74 m showed the lower *p*-value, *p* ≤ 0.1 (z-score = 1.3622).

For the Breteau Index (pupae), the cluster analysis by year did not show statistically significant patterns. However, the neighbourhoods with the highest BIs were Galán (BI = 18.18) in 2017, Prados del Norte (BI = 11.21) in 2018 and El Lago (BI = 12.5) in 2019. For the cumulative years (2017–2019), the highest BIs (BI > 8.33) were reported for the neighbourhoods Calle Nueva, Modelo, Libertador, Las Ferias and La Floresta. Table 4 shows a comparison of the cumulative values of pupae, the BIs and the cases reported during that period.

## 4. Discussion

This study modelled dengue data in the department of Cauca (Colombia) using spatio-temporal models, correlating the number of cases to environmental variables and estimating the relative risk at the municipality scale. Global and local measures of autocorrelation performed better than spatio-temporal measures when analysing the incidence rate of dengue. Dengue infection risk maps indicated the municipalities were above and below the reference risk, which allows the identification of how elevation and temperature can be used to estimate the probability of disease incidence.

### 4.1. Epidemiological Behaviour of High-Risk Clusters for Dengue in Cauca

Spatio-temporal clusters using the discrete Poisson model and the Getis Ord-Gi* spatial analysis technique have some similarities and differences. The identification of four clusters through retrospective spatio-temporal analysis limited the search for clusters with high rates to the municipalities of (1) Piamonte; (2) Patía and Sucre; (3) Miranda, Puerto Tejada, Villa Rica, Padilla, Corinto, Caloto, Santander de Quilichao and Guachené; and (4) López de Micay, Timbiquí, Suárez, El Tambo and Argelia. While this analysis allowed the relative risk of the clusters to be determined, the analysis of hot spots was not only consistent with the incidence rate patterns and identified the years when the rates were significantly high (≥90% confidence).

In 2018, Mutheneni et al. conducted a spatial analysis with Getis-Ord Gi*, as this was the method that showed the best autocorrelation [41]. The G Index of Getis and Ord [42] help to identify the degree to which the units of analysis with high values (hot spots) or low values (cold spots) are grouped; that is, it prioritized the formation of clusters.

The spatial distribution pattern of dengue cases was significantly clustered and identified dengue hot spots in the department of Cauca. A consistent dengue hot spot during the study period was identified in the municipality of Patía (≥95% confidence). The other significant hot spots (Miranda, Puerto Tejada and Villa Rica) were mainly located in the northeastern regions of the department near the Valle del Cauca, as Cali is the third most important city in Colombia and is a place with hyperendemic transmission of dengue [39]. The size of the region is related to the geographical characteristics of the territory, since the hot spots lie in highly transited areas along the Pan-American Highway, a commercial and human corridor that connects the departments of southwestern Colombia.

Regarding the space-time scan statistic and Getis-Ord technique, most proposed tests for spatial clustering are tests for global clustering. Some of those methods test for clustering throughout the study area without the ability to identify the location of specific clusters. As such, these tests and the spatial scan statistic complement each other, since they are useful for different purposes.

Additionally, it was observed that in the population of the municipality of Patía, the number of cases of dengue infection was slightly higher among men than among women. These results are similar to those of other studies [41,43,44,45] and contrast with those found in Molineros et al. and Restrepo et al. [46,47]. Additionally, the highest number of cases was found among those aged 27 to 59 years, followed by those aged 19 to 26 and 6 to 11 years. This could be because there was a higher number of cases reported in these age groups in this population. Additionally, the greater number of cases in these age groups may be due to the circulating dengue virus serotype and its mobility patterns.

### 4.2. Importance of the Environmental Variables Elevation and Minimum Temperature in the Prediction of Dengue

Elevation was the variable with the strongest relationship with the incidence of dengue (μ α_1_ = −0.02165). Mena et al. and Gyawali et al. (2021) showed that elevation is negatively associated with the incidence of dengue in Costa Rica and Nepal, respectively [48,49]. In Colombia, similar results have been reported. Vásquez found that altitude is a major environmental variable in the incidence of dengue, as increased altitude decreases the risk of dengue by between 60 and 89% within the altitudinal range in Cundinamarca, Colombia [50].

In Colombia, the vector Ae. aegypti has been reported at up to 2302 m.a.s.l. However, its infection with dengue virus has been indicated at a maximum height of 1984 m.a.s.l., in Bello, Antioquia [7]. In the municipality of Patía, the altitude ranges from 536 to 3264 m.a.s.l., and the areas with the highest probability of infection are located at El Bordo and El Estrecho, which is consistent with lower elevation values. Altitude has been indicated as a macroenvironmental factor that limits and influences the development of the vector and the virus [51]. Additionally, altitude is a modifying agent of the microclimate [50,52].

In this study, the minimum temperature was significantly associated with the incidence of dengue in the time window (μ α_2_ = −3.194), and 2016 and 2018 had minimum yearly temperatures of 17.68 and 17.62 °C, respectively. Other studies have associated the minimum temperature with a higher incidence of dengue. Tuladhar et al. found a greater correlation of dengue incidence with the minimum temperature than with the maximum temperature in Chitwan, Nepal [53]. Similar results were obtained in southern Taiwan [54]; in Mexico, where there is a rapid increase in risk when the average minimum temperatures rise above 18 °C [55]; and in Cali, Colombia, where a strong association has been established between minimum temperature and dengue outbreaks. These studies suggest that periods in which extreme daily temperatures are limited to the range of 18–32 °C promote the growth of the vector population, the amplification of the virus, increased vector feeding, and increased contact with the human host [56].

Carrington et al. performed a vector competence experiment that showed that temperature fluctuations between 18 °C and 20 °C promote more rapid dissemination of the dengue virus than constant temperatures of 20 °C, as only 18.9 days are required for 50% of the mosquitoes exposed to the virus to develop a disseminated infection [57]. These results indicate a greater potential for dengue virus transmission at low temperatures with natural fluctuations and an accelerated rate of dissemination. Although constant temperatures of 26 °C showed similar dissemination results, mortality at this temperature was higher than that at the fluctuating temperature. An increase to a temperature of 30 °C becomes harmful for the mosquito because such temperatures can increase mortality and affect the reproductive function of adults [58].

According to the climatic classification established by the Caldas–Lang climate zoning guidelines, the 50.9% of the municipality of Patía is located in the warm semi-humid climate, 17.4% in the temperate semi-humid zone, 12.7% in the warm semi-arid zone, 12.3% in the temperate wet zone, and the remaining percentage is distributed between cold wet, cold superhumid, and highly cold superhumid climates typical of altitudes above 2000 m.a.s.l. Based on these characteristics, it can be suggested that warm semi-arid and warm semi-humid climates, which are in altitude bands below 1000 m.a.s.l, are directly related to a high probability of dengue transmission. Additionally, the 125.6 km^2^ extension, in which the temperature ranges from a temperate semi-humid climate between 17.5 °C and 24 °C through the altitudinal strip 1000 to 2000 m.a.s.l., could promote dissemination of the virus more effectively. This confirms the altitudinal variation in temperature and the trends of dengue disease transmission in warm areas, which is limited even in temperate areas, as evidenced in the present study. Similarly, the probability of infection in 2016 and 2018 was related to the minimum temperature (17.68 and 17.62 °C), which was not necessarily the highest among the years studied, as would be expected in dengue transmission scenarios. However, it was consistent with the optimal temperature for vector competence.

Finally, although the relationship between precipitation and dengue was not significant, the minimum temperature, maximum temperature, and precipitation values recorded were consistent with the phases of El Niño and La Niña. The 2015 and 2019 years had the highest temperature values, consistent with the warm phase (28.51 and 29.23 °C, respectively), while 2017 had the highest average precipitation value (2160 mm), due to the cold phase, among other factors.

A study conducted in Mérida, Mexico, found that rainfall variation was the main variable that explained seasonal changes in the abundance of Ae. aegypti and in cases of dengue in this locality [59]. Similarly, precipitation is an environmental variable that has been related to an increase in dengue cases in different countries such as Puerto Rico, Thailand, and Venezuela [60,61]. Precipitation can influence the transmission of dengue through its impact on the vector population, since this variable generates a greater abundance of breeding sites and stimulates the hatching of eggs [60].

Regarding the El Niño Southern Oscillation (ENSO) climate phenomenon, which consists of a warm phase known as El Niño and a cold phase known as La Niña, data from the National Oceanic and Atmospheric Administration (NOAA) show that from 2015 to 2019, the year with the most intense and prolonged warm phase (El Niño) was 2015, which began in October 2014 and ended in April 2016 and was followed by another less intense warm phase from September 2018 to June 2019. Furthermore, the cold phases (La Niña) during this period were shorter and less intense. The most significant phase occurred between October 2017 and April 2018, while a less significant phase occurred between August and December 2016. The highest precipitation within the analysed time window was recorded in 2017.

The implementation of the spatial BYM model in estimating the RR resulted in significant differences of dengue transmission in Patía (Cauca). This analysis also offers the advantage of using count of disease cases when incidence is not so high. With this type of model, the residual relative risks in nearby areas are expected to be more similar than in faraway areas. Further research can be conducted by choosing the prior distribution with other parameter values such as sociodemographic or entomological values.

### 4.3. Relationship between the Dengue Disease in the Municipal Seat of Patía and the Spatial Distribution Pattern of the Vector

The distribution pattern disease does not appear to be significantly different than random. However, the neighbourhoods with the highest incidence rate corresponded to neighbourhoods with the highest number of accumulated dengue cases during the study period: Olaya Herrera, Libertador, Postobón, and Popular. The same was not true for the neighbourhoods Hueco Lindo and Calle Nueva, which, despite not having a significant number of cases, had a high density of cases due to the superposition of the values of the pixels (10 m × 10 m each), which is more noticeable in small and contiguous spaces such as those encompassed by these neighbourhoods.

It is important to mention that the neighbourhoods that composed the risk clusters identified using kernel density (see Appendix A) had a greater number of inhabitants than the neighbourhoods that did not represent any risk, which can affect the obtained results according to the provisions of Khormi and Kumar [62]. The Olaya Herrera neighbourhood features open spaces such as cemeteries and market squares, which could favour clustering and the presence of breeding sites, as could the neighbourhood’s proximity to rural areas. Other explanatory factors, such as access to basic sanitation services, water use and the disposal of water containers and the mobility of the population, should be considered in future studies.

Furthermore, the analysis of hot spots identified neighbourhoods with extreme and geographically homogeneous values in terms of the total number of pupae: Libertador, Olaya Herrera, and Modelo. These neighbourhoods had higher numbers of pupae than the other neighbourhoods; similarly, contiguous neighbourhoods had significant values, especially in the case of Libertador. One of the most important factors in this analysis was the critical distance of 100 m, which was defined according to the dispersal of the vector. Although this analysis focuses on the pupal stage, the threshold is consistent with other similar studies. Garelli et al. [63] found a clustering of pupae was found in a maximum radius of 150 m. The concentration of the high values found is of great relevance, since the spatial grouping of pupae is usually weak, as described by Khormi et al. and Garelli et al. [62,63]. In addition, this result may be associated with the presence of adult mosquitoes, as indicated by LaCon et al. [64].

Although we did not find a peak by incremental spatial autocorrelation, this most often happens in cases where data have been aggregated and the scale of the processes affecting the input field variable are smaller than the aggregation scheme.

When comparing the results of the analyses of the epidemiological and entomological data mentioned above, it is observed that the neighbourhoods of Libertador and Olaya Herrera simultaneously present the highest density of dengue cases and the highest pupae concentration. However, the correlation between the total number of cases and the total number of pupae registered in the time window of 2017–2019 was weak and non-linear, as demonstrated by the Spearman coefficient (0.1893).

### 4.4. Study Limitations

The findings of this study have to be seen in light of some limitations. First, in Phase 1, some of the cases were georeferenced using the neighbourhood centroid; we recommend to health authorities the use of tools that allow georeferencing the actual residence address of the case. Second, in Phase 2, the effect estimates in the model are based on the basis of a purely spatial model; even though some of the most important environmental variables were included, they could be improved further with entomological data. Third, in Phase 3, due to the fact that the surveillance of immature stages in the department of Cauca started in 2017, we did not use the same timeframe for epidemiological and entomological clusters.

An additional risk worth mentioning is that of security and public order in the department, as this may have an impact on the collection of primary information in the field. In addition, displacement in conflict zones can be a limitation for primary data collection, so it is recommended to have the support of stakeholders at the local level.

## 5. Conclusions

Previously, the dengue transmission risk stratification process consisted of the formation of socioecological and epidemiological strata, defined according to the distribution and frequency of risk factors and the endemic and hyperendemic transmission patterns at the local scale [8]. Currently, dengue risk stratification involves five operational scenarios that aim to explain and/or predict how environmental, sociodemographic, and entomological–epidemiological variables will increase or reduce the incidence of the disease [65].

This study is the first to report high-risk clusters of dengue for the department of Cauca based on spatial analysis techniques and considering the trends of dengue disease over a period of seven years. It also evaluated the influence of other variables, such as elevation, minimum temperature, maximum temperature, and precipitation over dengue behaviour in Patía, Cauca. Between the years 2015 and 2019, the main variables that were related to the presence of dengue were altitude and minimum temperature—findings that are consistent with the geographical and climatic conditions of the municipality. Although the effects of elevation on the incidence of dengue are widely known, the minimum temperature is emerging as an important variable, as it is positively related to temperature fluctuations and infection in the vector.

At the local scale, the study established an association between dengue conglomerates and the accumulation of *Ae. aegypti* pupae in the municipal seat of Patía. The neighbourhoods with the highest kernel density (Olaya Herrera, Libertador, Postobón and Popular) correspond to the neighbourhoods with the highest number of cumulative dengue cases during the study period. Similarly, the Libertador neighbourhood showed extreme and geographically homogeneous values in terms of the cumulative total pupae.

To determine the most cost-effective intervention strategy, it is necessary to construct operational scenarios. In this study, as a first step, the localities were stratified, taking the municipality of Patía as an example. In addition, information was obtained for environmental variables and entomological–epidemiological antecedents. Considering that scenarios should be based on the risks of transmission, the municipality of Patía is characterized as an area with high transmission of dengue virus within the department of Cauca. This is because this municipality has: (1) urban areas with a higher-than-average incidence in at least three of the last five years, according to the data generated by the epidemiological surveillance system; (2) there are established populations of the vector, (3) persistent transmission and (4) various outbreaks during the year, with seasonal behaviour, and (5) cases of severe dengue have been reported.

## Figures and Tables

**Figure 1 tropicalmed-08-00262-f001:**
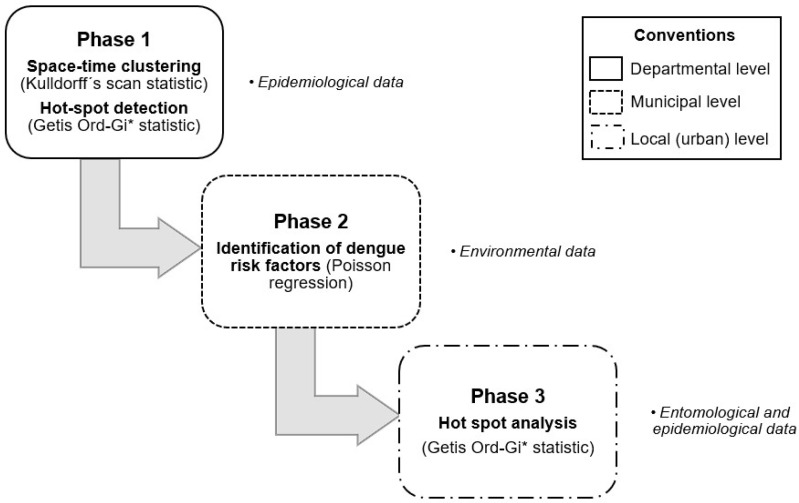
Methodological phases of the study design.

**Figure 2 tropicalmed-08-00262-f002:**
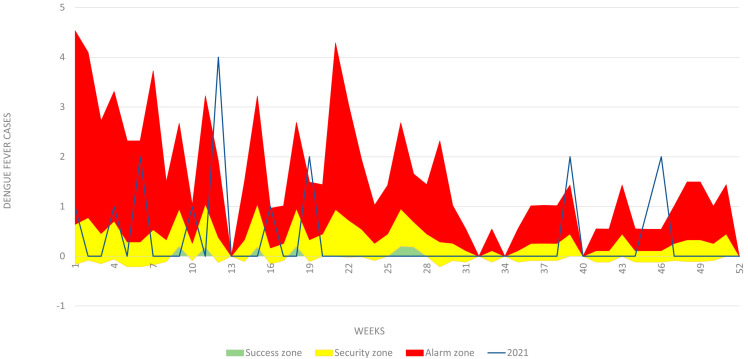
Dengue endemic channel for the municipality of Patía. Yellow zone represents the number of cases within the expected normal seasonal range. Anything above this moving threshold would be considered representative of an unprecedented number of cases in the municipality (Red zone).

**Figure 3 tropicalmed-08-00262-f003:**
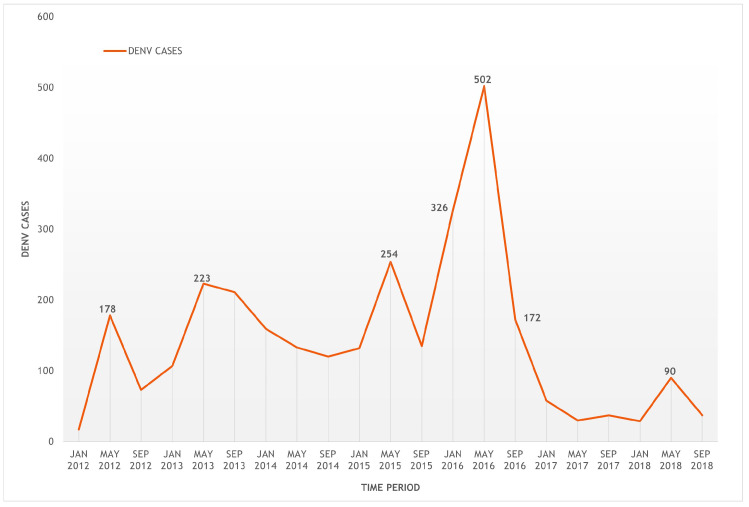
Time series of dengue cases from the department of Cauca from 2012–2018.

**Figure 4 tropicalmed-08-00262-f004:**
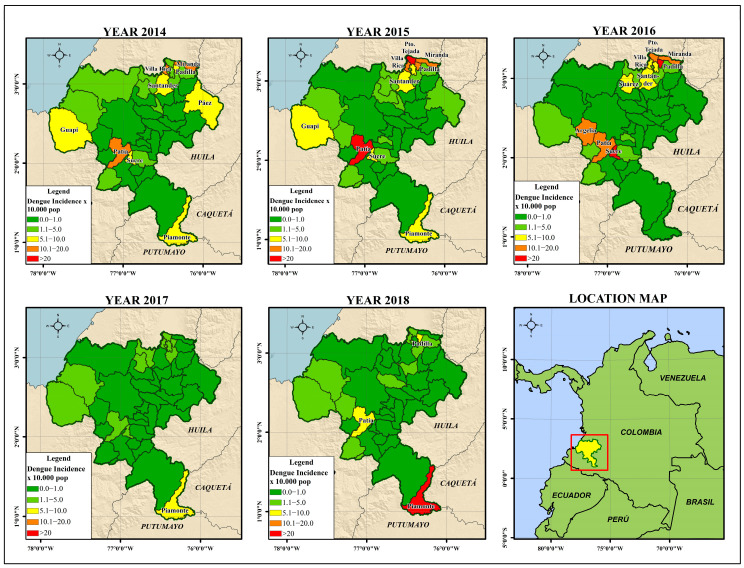
Dengue incidence rate × 10,000 population for the department of Cauca (2014–2018).

**Figure 5 tropicalmed-08-00262-f005:**
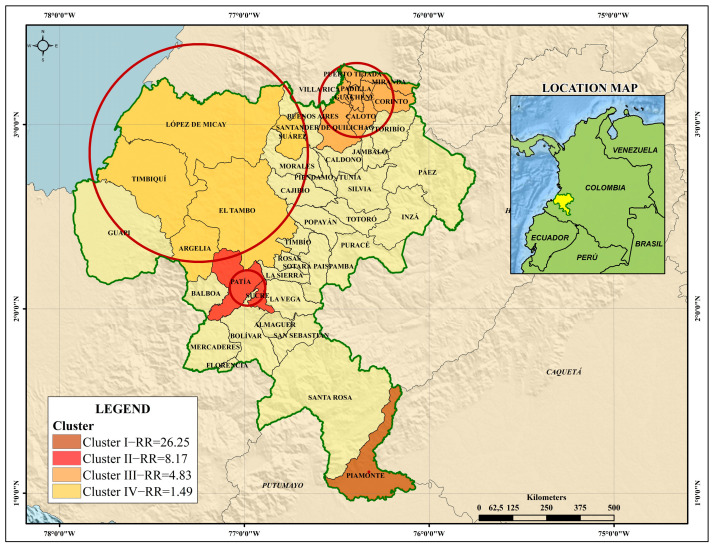
High-risk spatiotemporal clusters of dengue in the department of Cauca (2012–2018).

**Figure 6 tropicalmed-08-00262-f006:**
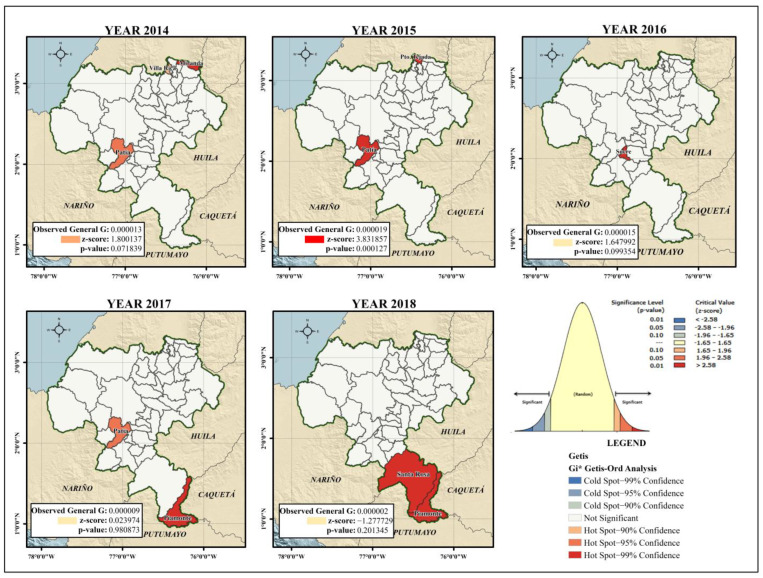
Hot spots for the department of Cauca considering the incidence rate of dengue (2014–2018).

**Figure 7 tropicalmed-08-00262-f007:**
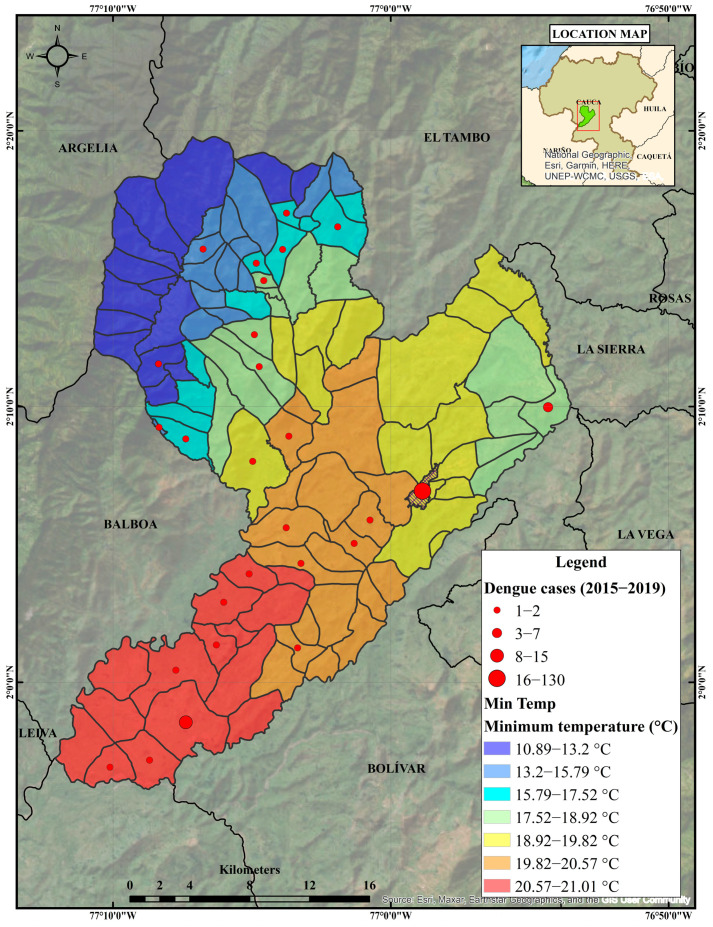
Observed number of dengue cases in Patía municipality for 2015–2019 period.

**Figure 8 tropicalmed-08-00262-f008:**
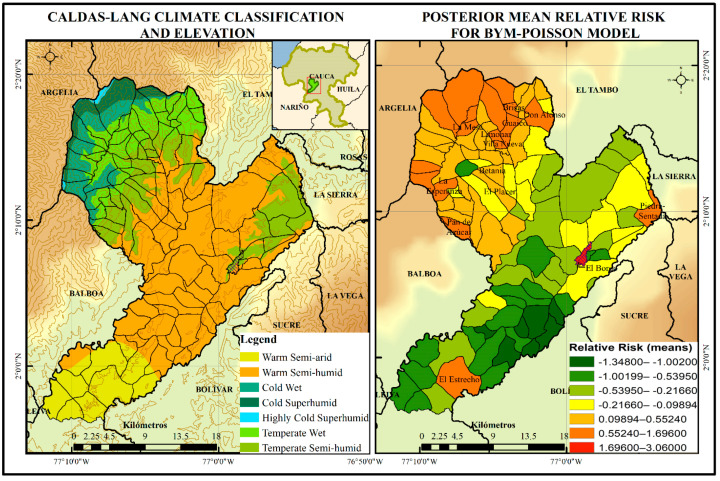
Poisson spatial BYM regression model for dengue disease based on environmental variables altitude and minimum temperature, years 2015–2019 (Patía, Cauca).

**Figure 9 tropicalmed-08-00262-f009:**
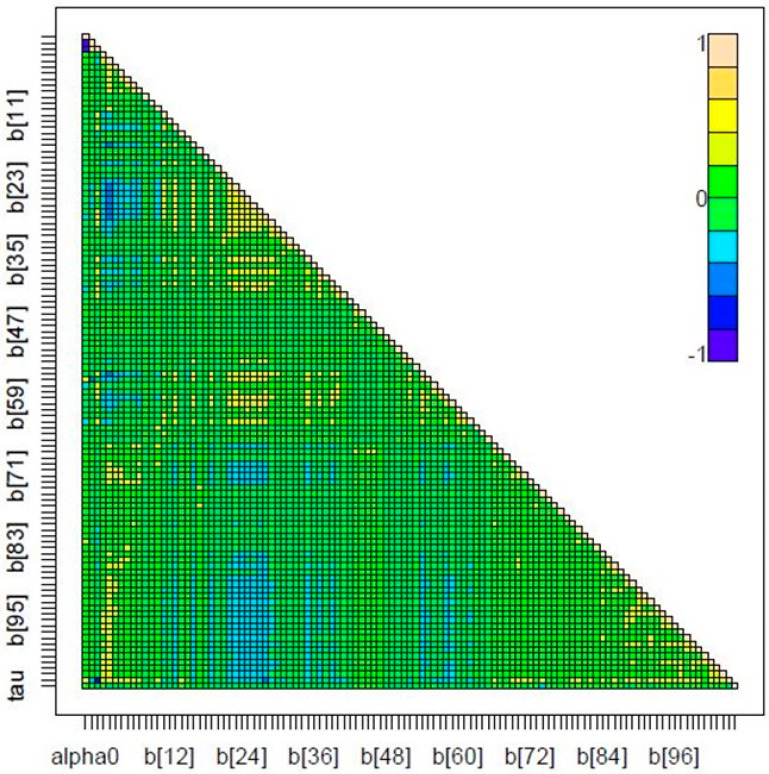
Cross-correlations for Markov chain in Poisson spatial regression model for dengue disease based on environmental variables of altitude and minimum temperature, years 2015–2019 (Patía, Cauca).

**Figure 10 tropicalmed-08-00262-f010:**
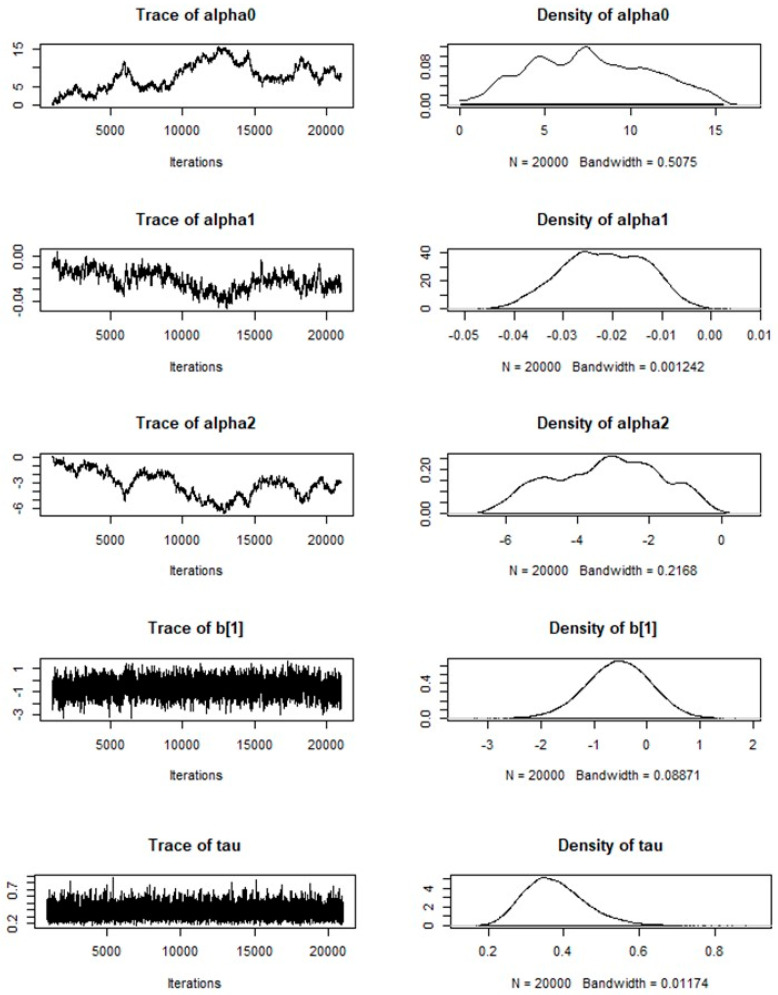
Trace and density for nodes in Markov chain for Poisson spatial regression model (2015–2019).

**Figure 11 tropicalmed-08-00262-f011:**
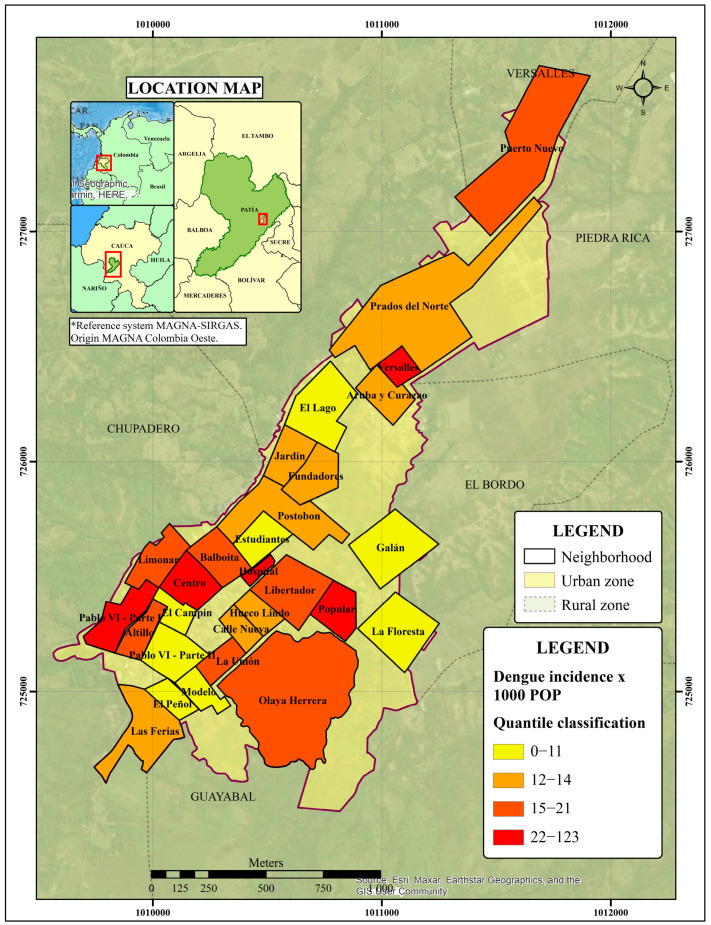
Dengue incidence rate per 1000 habitants in Bordo-Patía, Cauca (2015–2019).

**Figure 12 tropicalmed-08-00262-f012:**
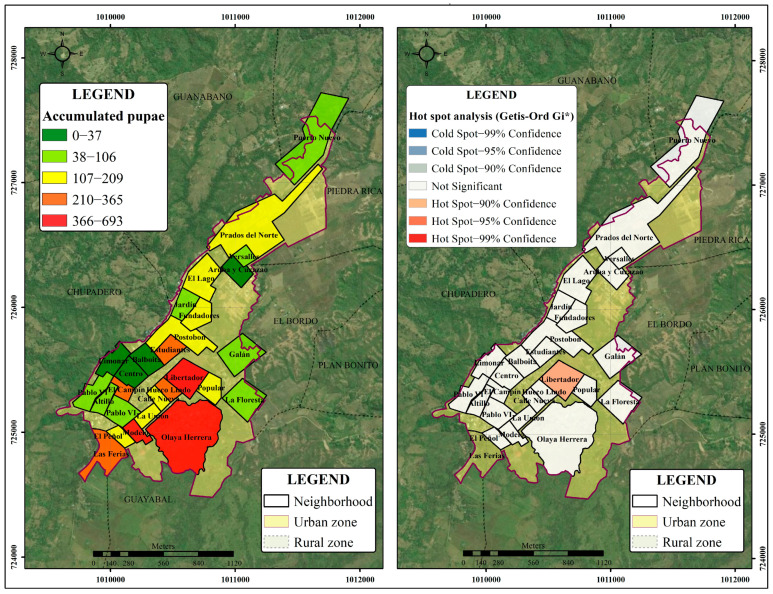
Accumulated pupae of A. aegypti (2017–2019) in the neighbourhoods of Bordo–Patía (**left**); hot spots for pupae calculated using the Getis-Ord Gi* statistic (**right**).

**Figure 13 tropicalmed-08-00262-f013:**
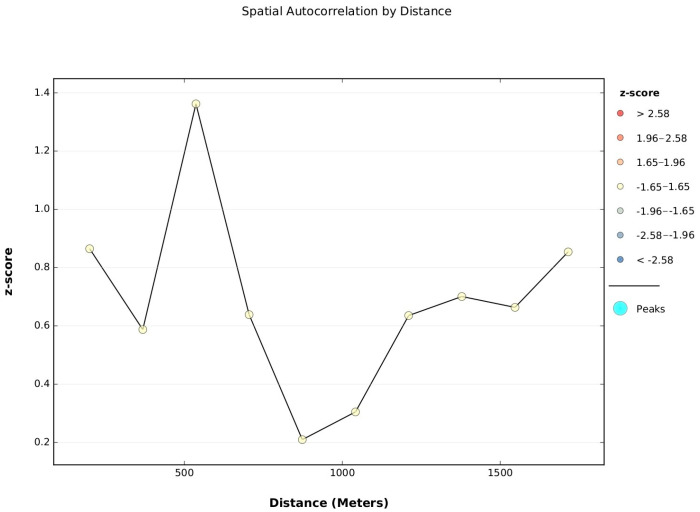
Incremental spatial autocorrelation for the accumulated pupae in Bordo–Patía neighbourhoods (2017–2019).

**Table 1 tropicalmed-08-00262-t001:** Spatio-temporal analysis for the identification of high-risk clusters for dengue in the department of Cauca (2012–2018).

Dengue Clusters in the Department of Cauca				
	Cluster I *	Cluster II *	Cluster III *	Cluster IV
**Municipalities**	Piamonte	Patía and Sucre	Caloto, Miranda, Corinto, Padilla, Puerto Tejada, Guachené, Santander de Quilichao and Villa Rica	López de Micay, Tim- biquí, Suárez, El Tambo and Argelia
**Time frame**	2018	2014–2016	2013–2015	2016
**Population**	7343	45,103	274,554	134,841
**Registered cases**	56	293	857	59
**Expected cases**	2.18	39.65	237.22	39.77
**Relative risk**	26.25	8.17	4.83	1.49
**Likelihood ratio**	128.58	345.14	566.61	4.11

******p* < 0.001.

**Table 2 tropicalmed-08-00262-t002:** Summary of the results of relative risk estimations using BYM model for dengue.

Node	Mean	Standard Deviation	Monte Carlo Standard Error	2.5%	Median	97.5%
alpha0	7.695	3.47	0.2902	1.795	7.505	14.39
alpha1	−0.02165	0.008491	6.66 × 10^−4^	−0.03802	−0.02159	−0.006483
alpha2	−3.194	1.483	0.1237	−5.948	−3.101	−0.6001
b [1]	−0.5435	0.6066	0.01957	−1.777	−0.5334	0.6039
b [2]	1.34	0.4015	0.01827	0.5572	1.337	2.132
b [3]	−0.5395	0.2034	0.01239	2.672	3.057	3.464
b [4]	−0.3136	0.5844	0.01519	−1.499	−0.3007	0.7992
b [5]	−0.1096	0.5187	0.01175	−1.165	−0.09905	0.8628
b [6]	−0.2194	0.5398	0.01347	−1.334	−0.2008	0.7841
b [7]	−0.572	0.5891	0.01551	−1.775	−0.5546	0.5265
b [8]	0.6091	0.628	0.01041	−0.7112	0.6408	1.763
b [9]	0.01779	0.5955	0.008791	−1.202	0.03556	1.121
b [10]	0.2088	0.5201	0.01189	−0.8448	0.2213	1.185
b [11]	0.3356	0.769	0.02283	−1.213	0.3575	1.799
b [12]	0.2252	0.5891	0.005614	−0.9738	0.2411	1.339
b [13]	0.6786	0.7378	0.02756	−0.8057	0.686	2.082
b [14]	−0.1665	0.5087	0.01047	−1.227	−0.1489	0.7755
b [15]	−0.3246	0.5874	0.01131	−1.532	−0.3095	0.7713
b [16]	0.5209	0.8719	0.03325	−1.256	0.5406	2.169
b [17]	−0.2656	0.8166	0.01115	−2.038	−0.2108	1.168
b [18]	0.2778	0.5413	0.004316	−0.8249	0.2923	1.291
b [19]	0.5524	0.9201	0.03325	−1.301	0.5725	2.294
b [20]	0.3277	0.5034	0.004141	−0.6865	0.3374	1.296
b [21]	0.5987	0.583	0.00776	−0.5936	0.6208	1.676
b [22]	0.6985	0.7948	0.02767	−0.9144	0.7118	2.201
b [23]	0.7003	0.7746	0.03044	−0.8398	0.7047	2.194
b [24]	0.6407	0.8443	0.03292	−1.052	0.6559	2.274
b [25]	0.6698	0.9354	0.03484	−1.199	0.6845	2.46
b [26]	0.6066	0.9397	0.03487	−1.268	0.633	2.4
b [27]	0.4907	0.8011	0.03154	−1.127	0.5083	2.01
b [28]	0.2674	0.7399	0.02962	−1.217	0.2898	1.654
b [29]	0.4655	0.7695	0.02498	−1.113	0.4954	1.902
b [30]	1.099	0.5708	0.01492	−0.0659	1.119	2.17
b [31]	0.5501	0.6691	0.01684	−0.813	0.5687	1.815
b [32]	0.4066	0.778	0.01396	−1.227	0.4386	1.83
b [33]	−0.0915	0.7589	0.7589	−1.718	−0.03475	1.237
b [34]	0.4784	0.5976	0.009939	−0.7641	0.4966	1.595
b [35]	0.849	0.7001	0.02512	−0.5624	0.8676	2.169
b [36]	0.9551	0.5268	0.005398	−0.1324	0.9757	1.924
b [37]	0.606	0.6236	0.02026	−0.6453	0.6139	1.809
b [38]	0.8178	0.5307	0.008372	−0.2679	0.8293	1.818
b [39]	0.4421	0.6346	0.02222	−0.8558	0.4562	1.663
b [40]	0.2733	0.6547	0.02012	−1.051	0.2935	1.513
b [41]	0.7322	0.5801	0.01306	−0.4673	0.7522	1.81
b [42]	0.3273	0.5643	0.006039	−0.8295	0.3457	1.381
b [43]	−0.341	0.706	0.01081	−1.842	−0.3008	0.9222
b [44]	0.05025	0.6627	0.006465	−1.334	0.07333	1.279
b [45]	−0.1234	0.6103	0.01085	−1.353	−0.1065	1.033
b [46]	−0.0973	0.5937	0.008167	−1.32	−0.07968	1.005
b [47]	−0.009787	0.5173	0.005345	−1.068	0.005371	0.9668
b [48]	0.1815	0.5157	0.00985	−0.861	0.1892	1.171
b [49]	0.1697	0.6075	0.005576	−1.071	0.1897	1.316
b [50]	0.3164	0.5631	0.006296	−0.8412	0.336	1.368
b [51]	0.09894	0.709	0.01129	−1.346	0.1211	1.418
b [52]	0.1512	0.6355	0.01506	−1.137	0.1628	1.357
b [53]	0.07944	0.5645	0.004516	−1.074	0.09702	1.143
b [54]	0.377	0.6513	0.02601	−0.9065	0.3821	1.643
b [55]	−0.6758	0.5308	0.0199	−1.748	−0.661	0.3264
b [56]	0.1952	0.5787	0.01601	−1.003	0.2111	1.279
b [57]	0.2195	0.6124	0.01644	−1.024	0.236	1.374
b [58]	0.4243	0.6111	0.02113	−0.7872	0.4305	1.605
b [59]	0.6599	0.834	0.02483	−1.075	0.6953	2.181
b [60]	0.1436	0.6062	0.01453	−1.076	0.1599	1.293
b [61]	0.1939	0.6635	0.01304	−1.185	0.2181	1.416
b [62]	−0.01998	0.7152	0.02696	−1.478	−0.01188	1.346
b [63]	0.2854	0.6176	0.009941	−0.9863	0.3086	1.432
b [64]	0.3602	0.6933	0.0105	−1.06	0.3934	1.629
b [65]	0.8406	0.7516	0.01541	−0.7825	0.9024	2.152
b [66]	−0.3531	0.5254	0.01345	−1.404	−0.3417	0.6417
b [67]	−0.3	0.6403	0.01245	−1.612	−0.2808	0.8996
b [68]	−0.2166	0.5635	0.0104	−1.381	−0.2001	0.8314
b [69]	−0.4795	0.6148	0.01522	−1.742	−0.4533	0.6633
b [70]	−0.7309	0.6564	0.01606	−2.078	−0.7153	0.5015
b [71]	−0.3847	0.6053	0.01633	−1.613	−0.3669	0.7566
b [72]	−0.2605	0.6073	0.0147	−1.523	−0.2339	0.8679
b [73]	−0.08215	0.6319	0.009134	−1.407	−0.04773	1.057
b [74]	1.696	0.4305	0.005831	0.802	1.714	2.492
b [75]	−0.0793	0.5685	0.009272	−1.267	−0.05448	0.9694
b [76]	−0.2972	0.6727	0.01103	−1.692	−0.2665	0.9358
b [77]	−0.04271	0.5972	0.009422	−1.281	−0.01751	1.048
b [78]	−0.2496	0.5543	0.00934	−1.413	−0.2206	0.7576
b [79]	−0.1282	0.5572	0.01277	−1.305	−0.1067	0.8917
b [80]	−0.4082	0.5174	0.01142	−1.498	−0.3859	0.5412
b [81]	−0.7311	0.6199	0.01116	−2.031	−0.6984	0.3958
b [82]	−0.7444	0.5425	0.01338	−1.871	−0.7151	0.2382
b [83]	−0.2832	0.9397	0.01321	−2.357	−0.1985	1.312
b [84]	−1.256	0.6806	0.01606	−2.659	−1.228	−0.009965
b [85]	−0.8898	0.655	0.01436	−2.256	−0.8608	0.3195
b [86]	−1.294	0.5551	0.01508	−2.444	−1.27	−0.27
b [87]	−1.348	0.7438	0.01577	−2.915	−1.31	0.007691
b [88]	−0.4158	0.6708	0.0184	−1.799	−0.3965	0.8323
b [89]	−0.05354	0.6186	0.01769	−1.314	−0.04162	1.119
b [90]	−1.061	0.5791	0.01582	−2.244	−1.046	0.03023
b [91]	−1.265	0.7097	0.01687	−2.74	−1.23	0.02105
b [92]	−0.8122	0.5955	0.01683	−2.022	−0.7983	0.3181
b [93]	−0.8615	0.6813	0.01718	−2.263	−0.8314	0.4095
b [94]	−0.8309	0.5883	0.01738	−2.03	−0.8136	0.2922
b [95]	−0.7791	0.6103	0.01846	−2.038	−0.76	0.3611
b [96]	−1.085	0.6701	0.01851	−2.488	−1.057	0.1453
b [97]	−0.2666	0.6203	0.01904	−1.56	−0.25	0.9118
b [98]	−1.002	0.7106	0.01762	−2.467	−0.976	0.3117
b [99]	−0.7158	0.573	0.01842	−1.878	−0.7047	0.3791
b [100]	−0.9311	0.6718	0.01815	−2.343	−0.8914	0.2785
b [101]	−0.5415	0.8362	0.01919	−2.312	−0.4927	0.9519
b [102]	−0.3021	0.6052	0.01899	−1.546	−0.2837	0.8592
b [103]	−0.5478	0.6663	0.0198	−1.923	−0.5185	0.6852
b [104]	−0.8285	0.7359	0.02025	−2.37	−0.7977	0.5246
b [105]	3.06	0.2034	0.01239	2.672	3.057	3.464
tau	0.3771	0.08271	0.001286	0.2407	0.3688	0.5622

**Table 3 tropicalmed-08-00262-t003:** Dengue cases registered in the municipality of Patía by source. Dengue cases from the ^a^ national and ^b^ department level.

Dengue Cases in the Municipality of Patia (2015–2019)					
	SIVIGILA Notified ^a^	CAUCA Notified ^b^	Provenience	Dismiss	Final Data
**Year**					
**2015**	101	97	109	12	97
**2016**	65	86	91	3	88
**2017**	14	14	23	9	14
**2018**	24	24	31	6	25
**2019**	17	22	28	6	22
**Total**	221	243	282	36	246

**Table 4 tropicalmed-08-00262-t004:** Dengue cases and Pupae Index in Bordo-Patía neighbourhoods, disaggregated by year (2017–2019).

Neighbourhoods	Population	Year 2017				Year 2018				Year 2019			
		No. Cases	Total Pupae	Breteau Index	Pupae/Person	No. Cases	Total Pupae	Breteau Index	Pupae/Person	No. Cases	Total Pupae	Breteau Index	Pupae/Person
Altillo	185	0	34	7.22	0.13	0	18	2.78	0.06	0	54	6.18	0.22
Aruba y Curazao	178	0	0	0.00	0.00	0	0	0.00	0.00	2	0	0.00	0.00
Balboita	365	0	9	2.38	0.02	0	0	0.00	0.00	1	17	2.08	0.04
Calle Nueva	291	0	67	3.57	0.12	0	35	1.92	0.10	0	20	0.69	0.02
Centro	263	0	2	1.59	0.01	0	0	0.00	0.00	0	35	4.17	0.15
El Campín	155	0	175	5.93	0.31	0	18	2.50	0.06	0	172	8.04	0.26
El Lago	523	0	9	4.17	0.04	0	12	7.14	0.09	1	102	12.50	0.28
El Peñol	416	0	42	5.36	0.10	0	33	5.41	0.12	0	60	4.25	0.12
Estudiantes	368	0	80	2.78	0.09	0	32	4.17	0.06	0	135	5.09	0.14
Fundadores	419	0	81	5.29	0.22	0	17	4.78	0.06	0	88	4.17	0.15
Galán	199	0	49	18.18	0.54	1	17.5	5.56	0.11	0	0	0.00	0.00
Hospital	8	0	0	0.00	0.00	0	0	0.00	0.00	0	0	0.00	0.00
Hueco Lindo	359	1	102.5	4.29	0.19	0	45	2.38	0.14	0	96	5.00	0.13
Jardín	303	0	63	1.85	0.09	0	0	0.00	0.00	0	29	2.78	0.03
La Floresta	281	0	43	15.38	0.41	0	12	7.14	0.10	0	0	0.00	0.00
La Unión	348	1	98	6.21	0.21	1	33	7.14	0.10	0	55	4.76	0.09
Las Ferias	86	0	163	14.29	0.49	0	15	2.78	0.07	0	108	9.44	0.21
Libertador	583	1	410	15.93	0.47	0	145	8.33	0.33	0	138	4.70	0.15
Limonar	163	0	0	0.00	0.00	0	0	0.00	0.00	0	14	2.50	0.05
Modelo	369	0	124	8.25	0.22	0	69	9.52	0.22	0	282	11.55	0.42
Olaya Herrera	1461	0	98	3.27	0.09	3	124.5	7.92	0.17	0	224	7.32	0.18
Pablo VI–Parte I	187	0	0	0.00	0.00	0	0	0.00	0.00	0	61	5.95	0.12
Pablo VI–Parte II	513	0	0	0.00	0.00	0	0	0.00	0.00	0	0	0.00	0.00
Popular	294	2	110	7.25	0.21	0	0	0.00	0.00	1	99	5.17	0.14
Postobon	717	0	114	6.23	0.16	2	15	1.43	0.03	0	60	3.66	0.07
Prados del Norte	327	1	33	2.38	0.12	0	42	11.21	0.20	1	77	5.14	0.14
Puerto Nuevo	198	0	60	5.81	0.23	0	0	0.00	0.00	0	21	1.96	0.06
Versalles	130	0	48	5.56	0.17	0	18	3.33	0.08	1	33	4.58	0.11

## Data Availability

Most of the relevant data are within the manuscript and its Appendix A. Additional data presented in this study are available on request from the corresponding author. Georeferenced data presented in this study are openly available in [Kaggle] at [Stratification of Dengue in Cauca—Colombia].

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
