# Peer review of "Spatial Analysis of Dengue Clusters at Department, Municipality and Local Scales in the Southwest of Colombia, 2014–2019"

_tropicalmed, 2023, doi:10.3390/tropicalmed8050262_

Round 1

Reviewer 1 Report

REVIEWER#1

General Comments:

The authors conducted a dengue epidemiology study in the department of Cauca, Colombia, to analyze the spatio-temporal distribution of dengue, considering environmental and entomological variables for the three different scales (department, municipality and local). Dengue data were analysed using ecological regression and clustering methods. The manuscript is relevant to the health public but some questions need more details.

Specific comments: - I suggest a new title: “Spatial analysis of dengue clusters at department, municipality and local scales in the Southwest of Colombia, 2014 – 2019”

- The relevant results and conclusions should be highlighted in the abstract.

- Keywords - Change “Aedes aegypti” to italics;

- I suggest that the authors provide a time series of dengue cases from the department of Cauca from 2012-2018;

- The authors should clarify, regarding phase 3: the term "local" can induce the reader to think about point-referenced spatial data, not area data. It would be appropriate to clarify If the analysis in phase 3 is local in the sense of considering neighborhoods within the municipality;

- For the epidemiological data: the methodology used was the same as for phase 1, differing only by the fact that in phase 1 it was considered department and in phase 3, it was considered municipality? Or in phase 1 spatial-temporal clustering was considered and in phase 3 a purely spatial cluster analysis was conducted?

- Could the authors include a map of the observed number of dengue cases and clarify in phase 2, which is the spatial unit of analysis (neighborhood)?

- Still in phase 2, clarify if a purely spatial (non-temporal) model was used. Was the model adopted for the total number of dengue cases for the period of analysis? How were the variables of temperature and precipitation included in the model of equation 1? Was the minimum temperature during the analysis period adopted for each neighborhood? Do these variables present relevant temporal variation? It is not clear to the reader how the authors dealt with the temporal dimension in the modelling of phase 2.

- The statement (Dengue incidence) in figure 10 is not in accordance with the image (Accumulated pupae);

- Why did the authors not use mosquito adult information? Table 5, it would be interesting if the authors could present the pupae disaggregated data of Ae. aegypti and Breteau index by year;

- Change the term “dispersion” to “dispersal”;

- I suggest that the authors include in the discussion section a paragraph on study limitations

Author Response

Thank you for your time and the review of our paper.

We have answered each of your points, please see the attachment.

Reviewer 2 Report

This is a well written manuscript, which is very important for the local monitoring and early warning of dengue fever. I recommend it to be published.

Author Response

Thank you for your review of our paper. 

We really appreciate your comments and find it very important to highlight their usefulness for local monitoring and early warning of dengue fever. This will be the next step of the study.

Reviewer 3 Report

This paper study the dengue which is an arbovirus transmitted by mosquitoes of the genus Aedes. But the research method is relatively simple. The transmission dynamics model should be added to simulate its transmission mechanism. In addition, the figures in the paper are not clear, and the format of the tables is also very confusing.

Author Response

Thank you very much in advance for your comments,

Reviewer 4 Report

Abstract

·       Brief problem statement was not found in abstract.

·       There data were given and described in results section however the key data were not stated.

Introduction

·       Problem statement of the study was not found.

·       What is the problem which lead to the need to study the Spatio-temporal distribution of dengue in the department of Cauca?

·       What is the problem that require the need to study the spatial stratification of dengue in the 83 municipality?

·       Does the problem relate to the need of action or efforts for management and control for dengue transmission using distribution data and analysis?

Methods

·       Figure 1 did not show the 3 phases described. Figure 1 need to be review and show where is the 3 phases.

·       Figure 2, there is no title for the figure. There is no description given about the 3 legends of the figure. There is a need to describe what is the meaning of success zone, security zone and alarm zone.

·       The spatio-temporal analysis was conducted to answer which objective?

·       Poisson regression analysis was conducted to answer which objective?

·       Spatial analysis for identification epidemiological and entomological clusters at the local scale was conducted for what purpose and to answer which objective?

Result

·       Figure 5 need to separate title and figure description. Description must be stated in the text of paragraph.

·       Why focus on municipality of Patia for analysis of spatial and environmental factors? Need to give reason.

·       Table 2, need to separate title and table description. Description must be stated in the text of paragraph.

·       Figure 7 and Figure 8, which lead to conclusion that the dengue cases are positively and are positively and spatially influenced by altitude and minimum temperature, did not show the data.

·       What are the values of altitude and temperature use for this conclusion? Where is the data?

·       Why choose Bordo-Patía, Cauca for disease and entomological cluster analysis? Please provide reason.

Discussion

·       For 4.2, value of elevation and temperature was stated and discussed, however there is no data of elevation and temperature were given and described in results section.

·       The discussion which described clustering of pupae was found in a maximum 836 radius of 150 m. However, there is no table or figure which show the relationship between pupae clustering and radius.

Author Response

Thank you for your review of our paper. 

Round 2

Reviewer 3 Report

Although the author has made improvements, they still does not use scientific methods or models to prove conclusions.

Author Response

Dear reviewer,

Reviewer 4 Report

Abstract

Problem Statement has been given, however the statement should be:

Dengue is an arbovirus transmitted by mosquitoes of the genus Aedes, where it has been one of the 15 main public health problems in the world as well as in Colombia. With limited financial, create a problem for management, hence there is a need for the municipality to prioritize target area for public health implementation. This study focus on the spatio-temporal analysis to determine the targeted area in managing the public health problems related to dengue cases.

Introduction

·       Problem statement of the study was not focus, it should be stated that the spatio- temporal analysis was conducted to assist in determining the problematic dengue areas to ensure an effective management of the dengue cases. This the key problem which lead to the need to study the Spatio-temporal distribution of dengue in the department of Cauca.

·       What is the problem that require the need to study the spatial stratification of dengue in the 83 municipality? This is not well described. It should clearly stated that with this study, the finding determine the level of risks, and help to determine action needed to control the risks.

Objectives of the study were stated, however the objective did not clearly state what is the problem it will help to solve. Hence suggestion was made as listed above.

Methods

·       The spatio-temporal analysis was conducted to answer which objective? No answer given for v2 which of the 3 objectives will this method answer to?

·       Poisson regression analysis was conducted to answer which objective? No answer given for V2, which of the 3 objectives will this method answer to?

·       Spatial analysis for identification epidemiological and entomological clusters at the local scale was conducted for what purpose and to answer which objective? which of the 3 objectives will this method answer to?

·       Which methods stated will help to provide answer for the main problem of the study, where the spatio-temporal analysis conducted will help to determine the targeted area in managing the public health problems related to dengue cases.

Result

·       Autor add one figure, which has change the figure numbering. Figure 5 has been change to Figure 6.

·       In Section 2.1: Study Area, authors should explain why focus on municipality of Patia for analysis of spatial and environmental factors. Need to give reason.

·       Figure 7 has been change to Figure 9, while Figure 8 has been change to Figure 10. ,

·       These two figures were used as findings which lead to conclusion that the dengue cases are positively and spatially influenced by altitude and minimum temperature. However, what are the values of altitude and temperature use for this conclusion? Where is the data?

·       Why choose Bordo-Patía, Cauca for disease and entomological cluster analysis? Please provide reason. Need to explain the reason in Section 2.1.

Discussion

·       For 4.2, value of elevation and temperature was stated and discussed, however there is no data of elevation and temperature were given and described in results section.

·       The discussion which described clustering of pupae was found in a maximum 836 radius of 150 m. However, there is no table or figure which show the relationship between pupae clustering and radius.

There are still many corrections need to be made. The authors try to provide too many data and technical results, which lead to difficulty to illustrate the need for the study to provide answers for the problem statement, which is spatio-temporal analysis conducted will help to determine the targeted area in managing the public health problems related to dengue cases.

Author Response

Dear reviewer, 
